# Proposal of New Natural Hydraulic Lime-Based Mortars for the Conservation of Historical Buildings

Marco Destefani *, Laura Falchi and Elisabetta Zendri

Department of Environmental Sciences, Informatics and Statistics (DAIS), Scientific Campus, Ca' Foscari University of Venice, 30170 Venezia, Italy; laura.falchi@unive.it (L.F.); elizen@unive.it (E.Z.)
* Correspondence: 858827@stud.unive.it; Tel.: +39-3808965178

**Abstract:** NHL mortars are known to be compatible materials for the conservation of architectural heritage. To improve their properties with regard to salt resistance and lower their carbon footprint, NHL-based mortars with salt inhibitor agents were studied and different formulations were produced: NHL-based mortars (MSs), composed of natural hydraulic lime; and sand and cocciopesto mortars (MSCs), in which NHL, sand and brick powder were admixed with two different products, diethylenetriaminapenta and chitosan, in different concentrations. The mortar performance was tested against freeze–thaw and salt crystallization through immersion–drying cycles in a 14% sodium sulfate solution. The results highlighted that the addition of cocciopesto was effective in increasing the salt resistance, but increased the water intake during the freeze–thaw tests. The use of DTPMP produced less thixotropic mortars and decreased the water uptake, but worsened the salt resistance of hardened mortars. Chitosan allowed a good workability of fresh mortar; its water uptake was similar to the reference mortar and slightly increased the salt resistance. In the cocciopesto samples, both additives reduced the weight variation during freeze–thaw tests; meanwhile, for the lime samples, the additives increased the weight variation during the final cycles.

**Keywords:** NHL-based mortars; green materials; restoration; durability; building conservation





## 1. Introduction

Concrete production is responsible for a negative impact on the environment, contributing 8%–10% of greenhouse gas emissions [1–3], 15% of global electrical energy consumption and the exhaustion of non-renewable resources [3]. It is, therefore, important to direct the construction sector towards a policy where conventional construction materials are replaced with by-products that will significantly reduce the environmental impact. Construction and demolition waste, glass waste, plastic waste, sludge from wastewater treatments and supplementary cementitious materials represent a starting point from which new construction materials can be designed [4–20]. Increasing the service life of a render product is another strategy to diminish the overall impact. Specific formulations need to be developed in relation to severe environmental conditions, e.g., salt resistance properties in coastal environments or freeze–thaw resistance in cold climates. A contribution to this can also come from the field of archaeological conservation, where the proposal of sustainable materials must be combined with the need for durable and compatible interventions [19–24].

In recent decades, a great amount of attention has been paid to proposing traditional mortars for the preservation of historical buildings and archaeological sites based on a reverse-engineering approach [25,26]. Among the traditional mortars, a mixture of limes with low-temperature fired-brick powder ("cocciopesto" mortar) has been demonstrated to be a possible durable and sustainable solution for obtaining hydraulic mortars [27–31]. Brick powder is a pozzolanic material capable of producing more flexible and permeable mortars [7–30]. Ca(OH)2 reacts with the pozzolanic material [32,33] to form stable

compounds, increasing durability and freeze–thaw resistance [34]. The freeze–thaw phenomenon can have dramatic consequences on infrastructures; it especially occurs in cold areas where ice forms. As the water freezes, the ice expands inside the cracks and repulsive forces push apart the stone, causing erosion and internal stress.

Less studied is the effect of brick powder addition to natural hydraulic limes (NHLs), even though this kind of mortar has been demonstrated to have adequate characteristics for new renders and plasters [35,36]. NHL-based mortars are more compatible with traditional masonries than cement-based mortars and are more durable than lime-based mortars. These characteristics make this kind of binder very interesting when preserving traditional buildings. The possibility of enhancing the positive mechanical and physical characteristics of NHL-based mortars with and without brick powder by the addition of water repellents or salt inhibitors is still an open and little-investigated issue.

In a coastal environment, the salt resistance of mortars, in addition to compatibility and sustainability characteristics, is currently a challenging goal [37–43]. Soluble salts present in salt water are able to penetrate porous building materials and crystallize within the pores, causing stress and internal damage. Moreover, marine aerosols deposit conspicuous amounts of salts on architectural surfaces that may retain humidity due to salt hygroscopicity and can penetrate within the material in the presence of free water. Usually, salt protection is achieved by preventing or reducing salt solution intake in porous materials or by applying water-repellent layers (e.g., silanes, polymeric layers, etc.) [44].

The addition of salt inhibitors has recently been proposed to enhance the durability of building materials (stone and bricks) to salt crystallization. Salt inhibitors are capable of modifying the crystalline morphology of the salt (e.g., ferrocyanide) or reducing the pressure between the growing crystal and the surface of the pore. The efficacy of inhibitors applied to building materials has been generally tested by mixing them in a soluble salt solution and evaluating the effects on the materials after the absorption of the solution and salt crystallization [45–48].

In recent years, the mixing of modifiers in a mortar during its production has been proposed to mitigate the salt crystallization effects as soon as the solution enters the mortar. The experimentation has generally been performed on lime-based and cement-based mortars [49–52].

Among others, two salt-inhibitor compounds are of increasing interest in enhancing the resistance of mortars: chitosan and diethylenetriamine penta(methylene phosphonic acid) (DTPMP). Chitosan is a linear polysaccharide, which acts by reducing the crystallization pressure between the growing crystal and the surface of the pore by forming a thin polymeric layer [48,50–54]. DTPMP is a stable phosphonic acid, which acts as a chelating agent for cations present in soluble salts; it is widely used in desalination plants to prevent scale formation and corrosion [55,56].

DTPMP and chitosan have been tested as possible inhibitors of salt crystallization in cement-based mortars [50,51,53–56], but, to the best of our knowledge, there are no studies on the performances of natural hydraulic lime (NHL) mortars with these compounds added. The aim of the present work was to evaluate the performance of compatible and sustainable mortars based on a natural hydraulic binder and natural hydraulic binder with the addition of brick powder to emulate the traditional brick-crushed ("cocciopesto") mortar. To enhance their resistance against salt crystallization, DTPMP and chitosan at were used different concentrations. The effects of these anti-salt agents as bulk admixtures were investigated. Special attention was paid to the evaluation of the rheology of the fresh mortars and the effects of the admixtures on the workability. Moreover, the behavior with respect to water was considered using capillary absorption and permeability determinations and a freeze–thaw resistance evaluation. To evaluate the performance of the anti-salt agents, salt-weathering tests were conducted by subjecting the specimens to absorption/drying cycles in a sodium sulfate solution [57,58]. All test results are represented through graphs created with the help of Origin 8.5 software in order to ensure easier interpretation of the results obtained.

## 2. Materials and Methods

Natural hydraulic lime mortar (MS series) and brick-crushed ("cocciopesto") mortar (MSC series) (average particles diameter 0.1μm) mockups with the addition of chitosan and DTPMP were produced. The natural hydraulic lime (NHL), white lime by "Lafarge", obtained through a calcination process at 1000/1100 °C of limestone containing about 10% of diffused silica [59], and the Ticino sand by "VAGA-Mapei Group", a limestone silica sand with a size fraction of 0.1 ÷ 0.9 mm, were selected. The cocciopesto used was obtained by crushing "San Marco Terreal S.p.A" red full brick with a size fraction of 0.1μm. Chitosan (medium molecular weight, technical grade) and DTPMP (diethylenetriaminapenta solution 50% (T) technical grade) were provided by "Sigma-Aldrich". The lime mortars were prepared by mixing NHL and sand with a mass ratio of 1:2, the cocciopesto mortars were prepared at a mass ratio of 1:1.7:0.3 NHL/sand/cocciopesto. The addition of the two admixtures was performed by replacing the mixing water with a 0.5%–0.25% aqueous solution of chitosan or DTPMP. Furthermore, mixtures with chitosan in powder form at 0.5 and 0.25% admixture/binder were prepared. A total of 350 mockups were produced, with 25 replicas for each formulation. Table 1 reports the names and composition of the different formulations.

**Table 1.** Sample composition of mockups. The lime: sand and brick powder ratio (in weight) is given in brackets.

| Sample | Composition |
| --- | --- |
| MS | Natural hydraulic lime NHL + Sand (1:2) |
| MSCL05 | Natural hydraulic lime NHL + Sand (1:2) + Chitosan (liquid) 0.5% |
| MSCL025 | Natural hydraulic lime NHL + Sand (1:2) + Chitosan (liquid) 0.25% |
| MSD05 | Natural hydraulic lime NHL + Sand (1:2) + DTPMP 0.5% |
| MSD025 | Natural hydraulic lime NHL + Sand (1:2) + DTPMP 0.25% |
| MSCS05 | Natural hydraulic lime NHL + Sand (1:2) + Chitosan (powder) 0.5% |
| MSCS025 | Natural hydraulic lime NHL + Sand (1: 2) + Chitosan (powder) 0.25% |
| MSC | Natural hydraulic lime NHL + Sand + Brick powder (1:1.7: 0.3) |
| MSCCL05 | Natural hydraulic lime NHL + Sand + Brick powder (1:1.7:0.3) + Chitosan (liquid) 0.5% |
| MSCCL025 | Natural hydraulic lime NHL + Sand + Brick powder (1:1.7:0.3) + Chitosan (liquid) 0.25% |
| MSCD05 | Natural hydraulic lime NHL + Sand + Brick powder (1:1.7:0.3) + DTPMP 0.5% |
| MSCD025 | Natural hydraulic lime NHL+ Sand + Brick powder (1:1.7:0.3) + DTPMP 0.25% |
| MSCCS05 | Natural hydraulic lime NHL + Sand + Brick powder (1:1.7:0.3) + Chitosan (powder) 0.5% |
| MSCCS025 | Natural hydraulic lime NHL + Sand + Brick powder (1:1.7:0.3) + Chitosan (powder) 0.25% |

The materials were mixed with tap water (w/b = 0.8) with a Mortar Mixer Paddle, 120 mm, triple helix-whip, threaded shM14 at a drilling speed of 100 rpm for a total of 3 min to allow a homogeneous distribution of different components. On the fresh mixtures, a flow test was performed following EN 1015-3 [60], using a Tecnotest manual flow table to determine the consistency of the mortar. Consistency was assessed by measuring the slump diameter obtained from the regulated amount of mortar on a shaking table. A total of three repetitions were carried out at t = 0, t = 30 and t = 60 min, respectively, to follow the variation in consistency over time. In addition, the viscosity of the fresh mixes was measured continuously from the right end of mixing using an IKA ROTAVISC hi-vi II

viscometer, with an SP-10 rotating arm set at 60 rpm, for a total duration of 15 minutes, in order to evaluate the dynamic viscosity of the mixes over time.

*Physical Behavior of the Hardened Samples*

The mixes were poured into $5 \times 5 \times 2$ cm silicone molds without demolding agents in order not to affect the surface properties. These mockups were stored at 70% RH for 48 h to promote the setting phase. The curing phase was then carried out in a controlled environment at 19 °C and 50% RH, for 28 days. To evaluate the physical behavior of the cured specimens, capillary absorption, permeability, compressive strength, freeze–thaw resistance and salt resistance were assessed. All tests were carried out with three replicates for each formulation, and average values for each point were reported in the tables. Water vapor permeability was evaluated following UNI-EN-1015/19 [61]; in particular, the WDD = density of moisture flow rate; $\mu$ = moisture resistance factor; $s_d$ = water vapor diffusion-equivalent air thickness were determined. Capillary water absorption of the samples was tested based on standard UNI-EN 1015/18 and UNI 10859 [62,63] at $20 \pm 1$ °C, the absorbed water was graphed against the square of time, then the capillary absorption coefficient CA was calculated as the slope of the initial part of the absorption graphs. The capillary index IC was evaluated by considering the ratio among the total water absorbed per surface unit over time and the total water*total time: $IC = \int f(Q_i)^* dt / (Q_{tf}^* t_f$ (where $Q_i$ = quantity of water absorbed per surface unit at time i; t = time, $t_f$ = final time. At the end of the absorption test, with fully impregnated mock-ups, it was possible to determine the total open porosity TOP as correspondent to the volume of water intruded by capillarity.

Compressive strength was tested on hardened mortars at 28 days according to EN 1015-11 [64].

In order to assess the frost resistance of the mortars, samples were subjected to freeze–thaw cycles according to EN 12371 [65]. The freeze–thaw cycles included a wetting period of three hours in which mockups were immersed in distilled water at room temperature, followed by a freezing step at $-20$ °C for 21 h. The salt resistance was tested according to EN 12370:2001 [66] with the following variations: Specimens were dried to constant weight at 20 °C prior to testing, then they were immersed in a 14% sodium sulphate decahydrate solution at ambient temperature ($20 \pm 5$ °C) for 2 h and dried at room temperature ($20 \pm 5$ °C) for 22 h instead of in an oven at 40 °C in order to simulate more realistic crystallization conditions (it is expected that a slower drying under standard environmental conditions would result in the formation of larger salt crystals in the mortar matrix, causing greater damage). Visual inspection was carried out after each cycle, while weight was monitored after each drying period to assess the salt solution absorption of salt-aged specimens. After reaching 15 cycles, the samples were desalinated by immersion in water at room temperature, and the salt extraction was monitored by measuring the electrical conductivity of the water. The total water volume was periodically replaced until constant electrical conductivity was achieved. After desalination, the samples were weighed, and the total weight variation was recorded.

After freeze–thaw cycles and the salt resistance tests, three replicates for each formulation underwent compressive strength test according to EN1015-11.

## 3. Results and Discussion

### 3.1. Results of Tests on Fresh Samples

Figure 1a,b shows the dynamic viscosity of NHL mortars (MS) (Figure 1a), NHL with brick powder (MSC) (Figure 1b) and with the different additives. In general, the viscosity of the samples decreased significantly in the 15 minutes of mixing, from about 3000 mPas/s to about 1000 mPas/s. The final values were quite similar for all samples. The NHL mortars show a more heterogeneous trend in viscosity values. The addition of DTPMP and chitosan increased the viscosity in the first minute of mixing, and then the viscosity decreased more than in the sample without additives. The presence of brick powder seems to determine a

more homogeneous behavior, and the influence of additives in the mortar formulation did not produce significant variations in the dynamic viscosity values (Figure 1b).

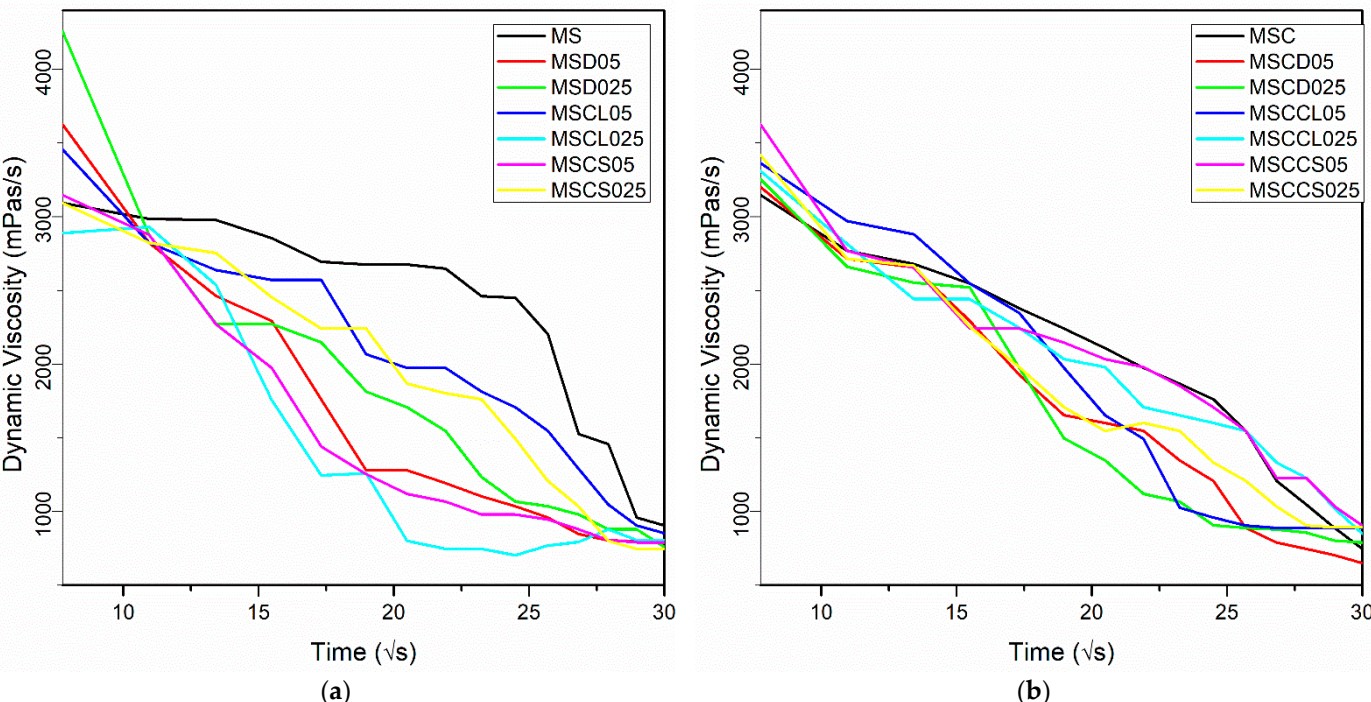

**Figure 1.** (**a**) Dynamic viscosity for NHL mortars (MS); (**b**) Dynamic viscosity for NHL with brick powder mortars (MSC).

The consistency of the different samples, expressed in terms of slump diameter (in mm), is shown in Table 2. Mortars based on NHL (MS) and NHL + brick powder (MSC) showed a similar consistency. The presence of DTPMP and chitosan in samples with NHL (MS series) had a different effect. DTPMP reduced the shrinkage to similar values independently of presence and amount of brick powder. The consistency of the mortars with added chitosan differs significantly: the samples produced with NHL show an increase in the slump diameter (239 and 226 mm), while samples with NHL and brick powder show values similar to those without chitosan (177 and 157 mm). The addition of chitosan to the powder has the same effect on the two series of samples (MS and MSC), with a slight reduction in consistency. The addition of DTPMP produces a light, not very thixotropic and soft mortar. This effect is more pronounced at the higher percentage (0.50%), especially in the MS sample series. Therefore, the MSD05 and MSD025 formulations do not seem to be suitable for vertical or inclined surfaces. On the other hand, chitosan produces mortars with a soft, homogeneous and thixotropic consistency, suitable for application on vertical surfaces.

**Table 2.** Flow test table measurements at 0, 30 and 60 min ($\Delta$ = average value in mm).

| Sample | Slump Diameter (mm) | | | $\Delta$(mm) |
| --- | --- | --- | --- | --- |
| | t = 0′ | t = 30′ | t = 60′ | |
| MS | 180 | 171 | 156 | 169 |
| MSD05 | 260 | 244 | 212 | 239 |
| MSD025 | 243 | 225 | 209 | 226 |
| MSCL05 | 160 | 142 | 126 | 143 |
| MSCL025 | 164 | 153 | 122 | 146 |
| MSCS05 | 175 | 158 | 145 | 159 |
| MSCS025 | 162 | 152 | 144 | 153 |
| MSC | 179 | 169 | 154 | 167 |

**Table 2.** *Cont.*

| Sample | Slump Diameter (mm) | | | Δ(mm) |
|---|---|---|---|---|
| | t = 0′ | t = 30′ | t = 60′ | |
| MSCD05 | 184 | 175 | 171 | 177 |
| MSCD025 | 164 | 157 | 149 | 157 |
| MSCCL05 | 150 | 147 | 143 | 147 |
| MSCCL025 | 160 | 153 | 145 | 153 |
| MSCCS05 | 166 | 156 | 145 | 156 |
| MSCCS025 | 165 | 152 | 141 | 153 |

*3.2. Hardened Mortars Characterization*

3.2.1. Hydric Behavior

Table 3 shows the apparent density, permeability characteristics and water absorption behavior of the mortars. Moreover, from the total absorbed water it was possible to determine the porosity of the specimens as TOP. Apparent density and total open porosity showed an inverse relationship, as expected. However, the DTPMP did not affect the porosity of MS samples, but reduced the porosity of MSC samples, while liquid chitosan slightly increased the porosity of both MS and MSC samples. Total open porosity and the permeability to water vapor are not perfectly correlated for each specimen; the differences are probably caused by a specific pore size distribution. It is possible that this difference also depends on an overall lower water absorption in the presence of certain additives. In particular, the use of DTPMP on MSC specimens caused a decrease in water absorption, whereas this behavior is not observed for MS specimens with similar ICs. Figure 2 highlights the fast water uptake of all mixtures with a saturation reached within 15 min since the beginning of the test. For MS specimens, the capillary absorption is more homogeneous between the different admixtures, while for MSC, an increased absorption is observed when chitosan is added (MSCCL05, MSCCS05, MSCCS025) Meanwhile, a decrease is observed when DTPMP is added (MSCD05 and slightly lower for MSCD025).

**Table 3.** Hydric behavior, data regarding water vapor permeability and capillary water absorption are listed: WDD = density of moisture flow rate; μ = moisture resistance factor; $s_d$ = water vapor diffusion-equivalent air thickness; IC = capillary index; CA = coefficient of water absorption by capillarity; TOP = total open porosity as determined by the volume of water absorbed by capillary absorption.

| Sample | | Apparent density | Water Vapor Permeability | | | | Capillary Water Absorption | |
|---|---|---|---|---|---|---|---|---|
| | | | TOP * | WDD | μ | sd | CA | IC |
| | | g/cm$^3$ | % | g/(m$^2$·s) | | m | mg/cm$^2$*s1/2 | |
| MS | mean | 1.71 | 16.6 | 58.16 | 14.23 | 0.30 | 2.33 | 1.09 |
| | σ | 0.03 | 0.6 | 3.24 | 0.87 | 0.01 | 0.60 | 0.23 |
| MSD05 | mean | 1.64 | 17.7 | 59.64 | 13.81 | 0.30 | 2.50 | 1.19 |
| | σ | 0.05 | 0.2 | 0.55 | 0.14 | 0.003 | 0.11 | 0.03 |
| MSD025 | mean | 1.68 | 15.6 | 57.16 | 14.48 | 0.31 | 2.33 | 1.32 |
| | σ | 0.06 | 0.3 | 1.86 | 0.53 | 0.01 | 0.16 | 0.13 |
| MSCL05 | mean | 1.61 | 19.0 | 53.13 | 15.69 | 0.34 | 3.44 | 1.17 |
| | σ | 0.03 | 0.4 | 1.87 | 0.61 | 0.01 | 0.35 | 0.07 |
| MSCL025 | mean | 1.74 | 14.7 | 58.51 | 14.11 | 0.30 | 2.28 | 1.23 |
| | σ | 0.04 | 0 | 0.37 | 0.1 | 0.002 | 0.12 | 0.11 |
| MSCS05 | mean | 1.69 | 16.5 | 54.10 | 15.42 | 0.33 | 2.17 | 1.31 |
| | σ | 0.01 | 0.3 | 3.42 | 1.03 | 0.02 | 0.09 | 0.14 |
| MSCS025 | mean | 1.72 | 17.4 | 74.43 | 10.81 | 0.23 | 2.44 | 1.21 |
| | σ | 0.03 | 0.3 | 3.45 | 0.55 | 0.01 | 0.43 | 0.05 |
| MSC | mean | 1.62 | 21.8 | 52.29 | 15.96 | 0.34 | 2.20 | 0.71 |
| | σ | 0.18 | 0.9 | 0.70 | 0.23 | 0.004 | 0.45 | 0.01 |
| MSCD05 | mean | 1.62 | 17.4 | 59.08 | 15.55 | 0.33 | 2.43 | 0.90 |
| | σ | 0.13 | 0.7 | 25.55 | 7.33 | 0.16 | 0.19 | 0.17 |

**Table 3.** *Cont.*

| Sample | | Apparent density | Water Vapor Permeability | | | | Capillary Water Absorption | |
| | | | TOP * | WDD | μ | sd | CA | IC |
| --- | --- | --- | --- | --- | --- | --- | --- | --- |
| | | g/cm$^3$ | % | g/(m$^2$·s) | | m | mg/cm$^2$*s1/2 | |
| MSCD025 | mean | 1.65 | 16.9 | 72.60 | 11.18 | 0.24 | 2.44 | 1.15 |
| | σ | 0.07 | 2.7 | 8.18 | 1.42 | 0.03 | 0.19 | 0.24 |
| MSCCL05 | mean | 1.55 | 21.1 | 61.35 | 13.95 | 0.30 | 2.17 | 0.95 |
| | σ | 0.05 | 0.6 | 16.55 | 4.14 | 0.09 | 0.09 | 0.10 |
| MSCCL025 | mean | 1.51 | 23.3 | 77.48 | 10.49 | 0.23 | 2.74 | 0.95 |
| | σ | 0.07 | 0.8 | 13.43 | 2.06 | 0.04 | 0.66 | 0.11 |
| MSCCS05 | mean | 1.54 | 23.3 | 48.34 | 17.37 | 0.38 | 3.43 | 0.96 |
| | σ | 0.11 | 4.4 | 0.65 | 0.25 | 0.01 | 0.36 | 0.15 |
| MSCCS025 | mean | 1.63 | 19.9 | 40.58 | 21.05 | 0.45 | 2.28 | 1.01 |
| | σ | 0.02 | 0.5 | 3.50 | 1.94 | 0.041 | 0.12 | 0.04 |

\* TOP was estimated by evaluating the volume of water absorbed at the end of the capillary test.

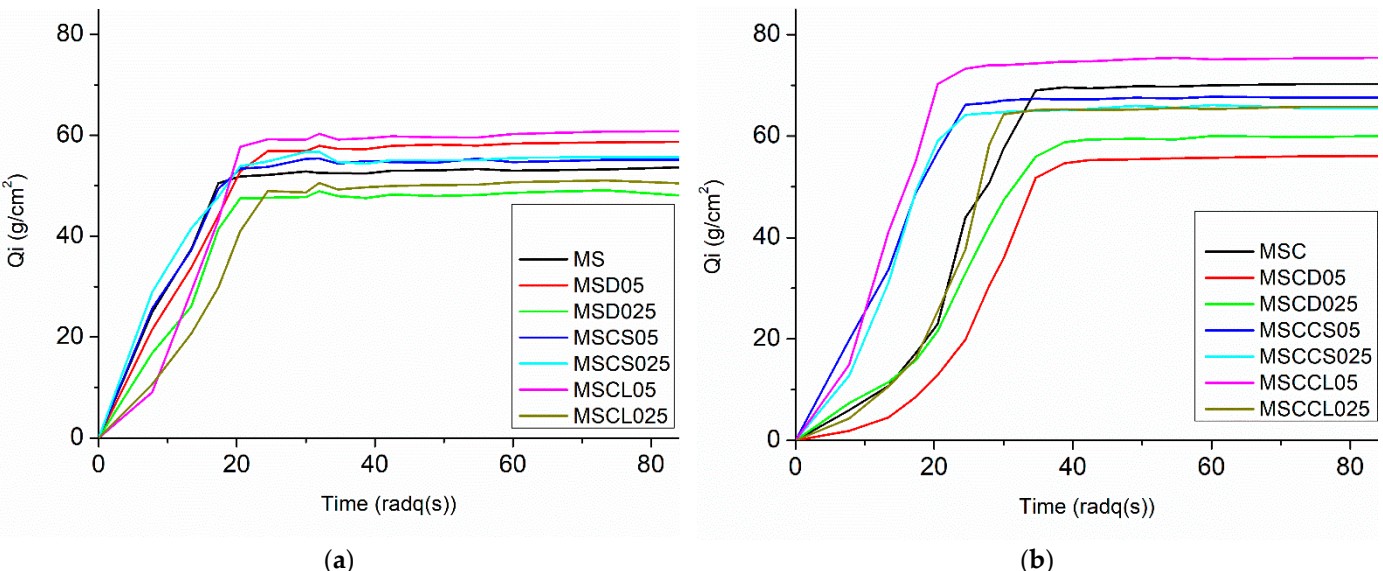

**Figure 2.** (**a**) Capillary absorption for MS sample series; (**b**) Capillary absorption for MSC sample series. Qi expresses the quantity of water absorbed per surface unit.

### 3.2.2. Determination of Frost Resistance

The weight variation in the MS samples during the cycles was characterized by a slight weight reduction in the first cycles followed by an almost constant weight (Figure 3). During the cycles, different trends were observed for admixed and non-admixed samples, especially in the presence of cocciopesto. MSC showed an increase in weight variation, possibly due to an increase in open porosity and a higher water retention rate. The lower weight variation observed with the added admixtures suggests that there is less variation in porosity and that the admixtures are effective at protecting the specimen from freeze–thaw, in particular MSCCL05, MSCD05 and MSCCS025 when the higher amount of additive is present in the mixture (0.5%). All MS mixtures showed negligible weight variation, but again MS showed a higher weight variation compared to admixed specimens. The presence of additives can mitigate, albeit partially, frost damage in the case of brick powder samples. The MS samples, on the other hand, showed a more uniform weight variation in the added mortars. At the end of the freeze–thaw cycles, most of the samples appear to be slightly crushed, without major damage. In the case of MSD05 and MSCD05 samples, the upper layer came off, as shown in Figure 4.

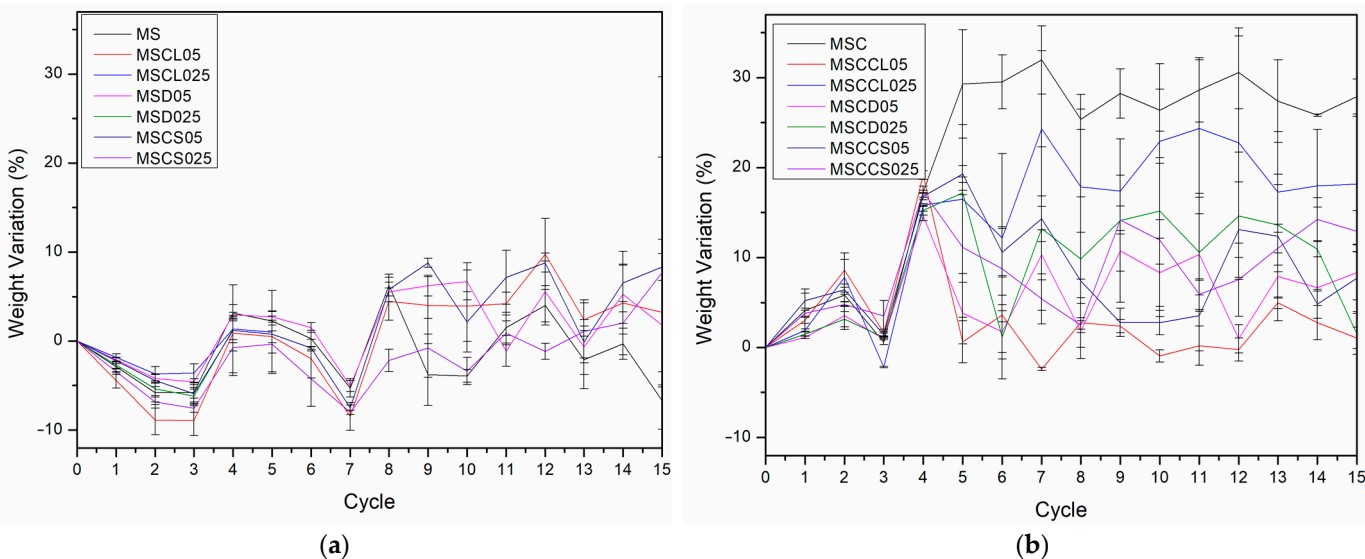

**Figure 3.** (**a**) Weight variation after each freeze–thaw cycle for MS sample series; (**b**) Weight variation after each freeze–thaw cycle for MSC sample series.

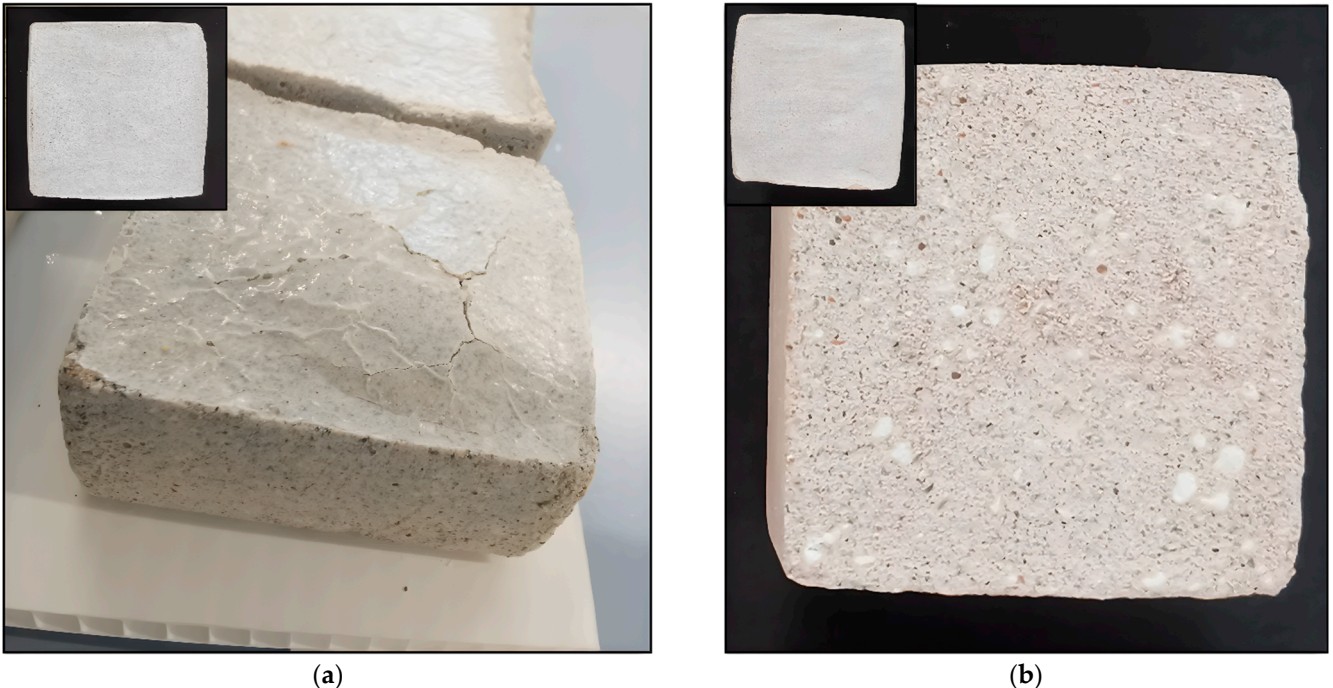

**Figure 4.** (**a**) MSD05 sample before (top left-hand corner) and after undergoing freeze–thaw cycles; (**b**) MSCD05 sample before (top left-hand corner) and after undergoing freeze–thaw cycles.

### 3.2.3. Determination of Salt Resistance

Table 4 reports the overall weight difference after sample desalination, and Figure 5 reports the weight variations during the salt crystallization test.

During the first cycles, the MS mortar samples show greater resistance to salt cycles. The weight variation for the first cycles proves to be almost zero, in contrast to the samples with brick powder component, which seems to increase the absorption and retention of water and salts (Figure 5a,b). Not all samples made it to the end of the 14 + 1 cycles; MSD025 and MSCL025 were severely damaged during cycles 6° and 8°, respectively, preventing further analysis (Figure 5a). At the end of the cycles and after desalination, the total weight

variation of the MS samples was higher than that of the MSC samples. In addition, samples with added DTPMP and chitosan (liquid) at 0.50% showed less weight loss.

**Table 4.** Overall weight variation at the end of the test for resistance to salt crystallization.

| Sample | Weight Variation (%) |
|---|---|
| MS | −18.48% |
| MSCL05 | −16.73% |
| MSCL025 | / |
| MSD05 | −9.76% |
| MSD025 | / |
| MSCS05 | −10.34% |
| MSCS025 | −14.03% |
| MSC | −7.92% |
| MSCCL05 | −0.10% |
| MSCCL025 | −4.02% |
| MSCD05 | −1.10% |
| MSCD025 | −4.99% |
| MSCCS05 | −6.08% |
| MSCCS025 | −4.94% |

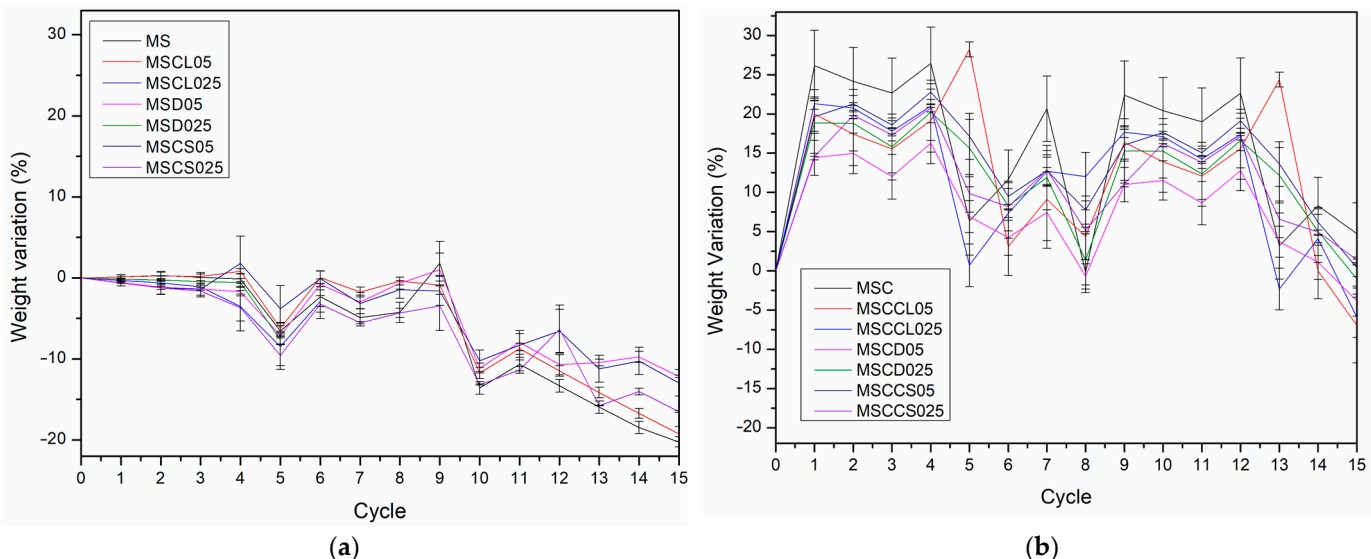

(**a**)      (**b**)

**Figure 5.** (**a**) Weight variation during salt crystallization test cycles of MS samples; (**b**) Weight variation during salt crystallization test cycles of MSC samples.

Salt-induced decay, based on a variation of the standardized method, determined the growth of efflorescences around each sample. The appearance of the specimens did not show any difference between the pure and added ones, but the slightest touch causes the effloresces to fall together with some mortar powder (Figure 6). Although they show a tendency to lose mass due to salt crystallization, their general appearance is less damaged than that of the samples with added brick powder (Figures S1–S4). Likely due to their higher water absorption, these samples are better able to retain the salt solution, limiting the weight loss of the damaged matrix.

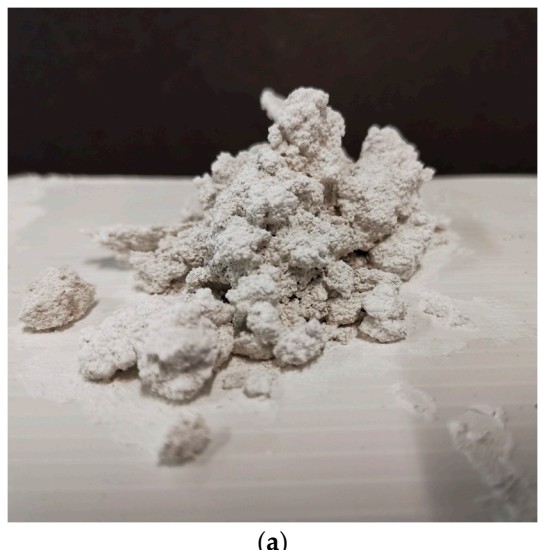

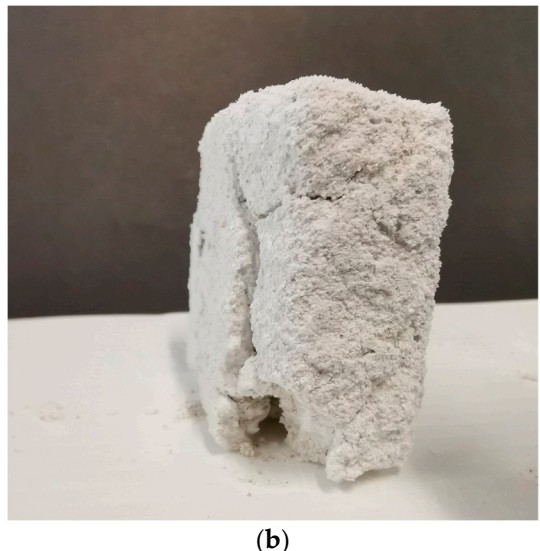

(**a**)                                                    (**b**)

**Figure 6.** (**a**) MSCL025 sample after 6° saline cycle; (**b**) MSD025 sample after 8° saline cycle.

### 3.2.4. Determination of Resistance to Compression of Hardened and Weathered Mortars

In order to evaluate the performances under compression of the mockups, tests were carried out by crushing mockups before and after salt crystallization and freeze–thaw cycles.

As Figure 7 shows, when comparing the compressive strength after 28 days and the water vapor permeability, an inverse relationship was observed. Thus, a higher porosity leads to lower compressive strength for both MS and MSC specimens independently of the admixture used. A slight positive effect of chitosan on the compressive strength of MS specimens was observed, possibly due to a higher moisture retention promoting a better NHL hydration (chitosan is a hygroscopic molecule able to retain water). A similar effect was observed when liquid chitosan was used in MSC. In MSC mixtures, the presence of DTPMP increases the density of the specimen and thus the compressive strength.

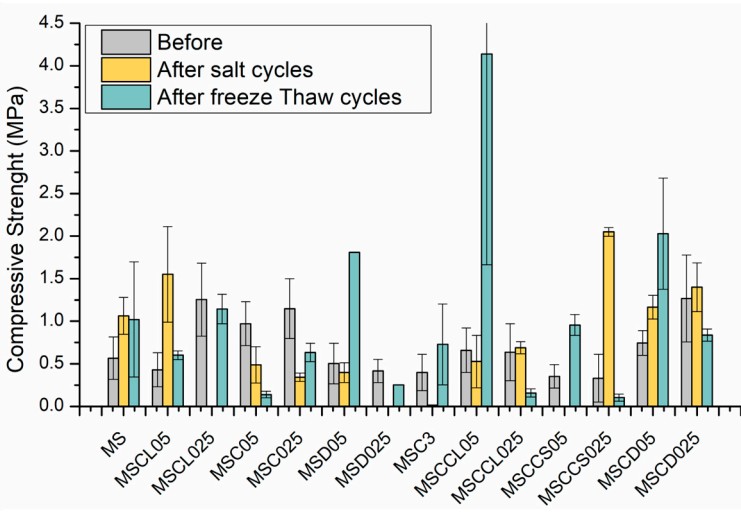

**Figure 7.** Compressive strength compared between undamaged, post-salt-crystallization and post-freeze–thaw-cycle samples.

The increase in compressive strength observed for MS and MSC samples after freeze–thaw cycles is attributable to a further hydration of the mortar and a pore structure able to withstand freezing cycles. The use of chitosan seems to slightly increase the resistance, with the exception of MSCS05. DTPMP guarantees compressive strength through MS and MSC samples.

In many cases, the specimens were unable to withstand the salt cycles and a general reduction in compressive strength was observed, with the notable exception of DTPMP in MSC specimens. Further hydration leading to higher compressive strength does not counterbalance the damaging effect of salt crystallization.

## 4. Conclusions

This study is the first to investigate the possibility of improving the performance and durability of NHL-based mortars in the presence of soluble salts and against freeze–thaw events by adding DTPMP and chitosan at different concentrations.

Chitosan has recently been re-evaluated and studied as a carbonation catalyst for lime mortars, while DTPMP, although generally used as a desalination treatment, has undergone various studies on its effect as a retarder in cement mortars. Currently, few sources have been found on the incorporation of these additives into natural hydraulic lime mortars, hence the purpose of our study. Special attention was paid to the evaluation of the rheology of the fresh mortars and the effects of the admixtures on the workability. Moreover, the behavior towards water was studied by capillary absorption and permeability determination.

Two sets of samples were prepared, based on NHL binder and NHL binder with brick powder. The structures based on cocciopesto generally exhibit improved durability and performance compared to pure lime mortars, approaching the most recent cement mortars' physical and chemical properties [31,34,35].

To evaluate the performance of the anti-salt agents, DTPMP and chitosan, salt weathering tests were carried out by subjecting the specimens to absorption/drying cycles in a sodium sulfate solution. Freeze–thaw resistance was also evaluated.

From a rheological point of view, DTPMP generally reduces the consistency of NHL-based mortars, producing soft and low thixotropic mixtures that are not always suitable for vertical or inclined surfaces. Chitosan produces mortars with a soft, homogeneous, and thixotropic consistency, suitable for application on vertical surfaces. Chitosan behaves differently depending on the composition of the samples by increasing consistency of the mortar in the presence of brick powder and decreasing it in samples without brick powder.

DTPMP is able to impart a partial hydrophobic behavior to the mortar, reducing water absorption by capillarity in both the series of samples. The resistance to salt crystallization is increased by the presence of brick powder, and, for these samples, the presence of DTPMP seems to make a positive contribution. Exceptions were the NHL samples with 0.5% of DTPMP and with 0.5% chitosan added, which proved to be not durable under the salt crystallization.

The freeze–thaw resistance does not seem to be increased by the presence of DTPMP and chitosan.

It is still difficult to predict the durability of the mortars against salt attack or thawing of mortars on the basis of their compositional, physical and chemical characteristics. Nevertheless, we have been able to provide, albeit loosely, an overview of the performance of different types of innovative mortars mixed with low environmental impactful products, of which the presence of pozzolanic mortars was the most prominent. Future investigations on these materials would clarify the dynamics of mortar hydration and the kinetics of the evolution of the chemical and physical properties of the formulations in presence of additives.

**Supplementary Materials:** The following supporting information can be downloaded at: https://www.mdpi.com/article/10.3390/coatings13081418/s1, Figure S1: MS series samples (part 1) before (left) and after (right) 14 saline cycles; Figure S2: MS series (part 2) samples before (left) and after (right) 14 saline cycles; Figure S3: MSC series samples (part 1) before (left) and after (right) 14 saline cycles; Figure S4: MSC series (part 2) samples before (left) and after (right) 14 saline cycles.

**Author Contributions:** Conceptualization, E.Z. and M.D.; Methodology, E.Z., L.F. and M.D.; Investigation, M.D. and L.F.; Resources, E.Z.; Data Curation, E.Z., L.F. and M.D.; Writing—Original Draft Preparation, M.D.; Writing—Review and Editing, E.Z., L.F. and M.D.; Supervision, E.Z. and L.F.; Funding Acquisition, E.Z. All authors have read and agreed to the published version of the manuscript.

**Funding:** This research received no external funding.

**Institutional Review Board Statement:** Not applicable.

**Informed Consent Statement:** Not applicable.

**Data Availability Statement:** Data is unavailable due to privacy restrictions.

**Conflicts of Interest:** The authors declare no conflict of interest.

## Abbreviations

| | |
|---|---|
| NHL | natural hydraulic lime mortars as defined by UNI EN 459-1:2015Building lime—Part 1: Definitions, specifications and conformity criteria [67] |
| MS | NHL Mortar Samples |
| MSC | Cocciopesto (minced bricks) -NHL Mortar Samples |
| C | Chitosan, specifically CL= chitosan in liquid solution at 0.5% or 0.25%, and CS= chitosan as solid powder |
| D | DTPMP, diethylenetriamine penta(methylene phosphonic acid) |
| 05 | 0.5%. by binder weight |
| 025 | 0.25%. by binder weight |
| RH | relative humidity |
| WDD | density of moisture flow rate according to UNI-EN-1015/19 [61] |
| $\mu$ | moisture resistance factor according to UNI-EN-1015/19 [61] |
| $s_d$ | water vapor diffusion-equivalent air thickness according to UNI-EN-1015/19 [61] |
| CA | capillary absorption coefficient according to UNI-EN 1015/18 and UNI 10859 [62,63] |
| IC | capillary index IC according to UNI-EN 1015/18 and UNI 10859 [62,63], IC = $\int f(Q_i)*dt/(Q_{tf}*t_f)$ where $Q_i$= quantity of water absorbed per surface unit at time i; t = time, $t_f$ = final time. |
| TOP | total open porosity determined by the water volume intruded according to UNI-EN 1015/18 [62] |

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
