# Peer review of "Proposal of New Natural Hydraulic Lime-Based Mortars for the Conservation of Historical Buildings"

_coatings, doi:10.3390/coatings13081418_

Round 1

Reviewer 1 Report

The authors have presented a research paper with very little depth in the essays and writing. Unfortunately, I cannot recommend this paper for publication in a scientific journal. I attach some comments justifying the Reject.

Line 16 and others, use dots "." as decimal separator.

In the abstract it would be convenient to include some significant numerical results.

The introduction is very poor, the auotres mention a large number of papers but do not discuss them. It is not a matter of including 56 references in 30 lines, it is a matter of carefully reviewing the literature, highlighting the relevant points of the other papers and highlighting the novelty of the proposed research. This point should be significantly improved.

It is recommended that a brief description of the structure and organisation of the paper be included after the objective of the paper.

In section 2. Materials and Methods, an introductory section is needed, followed by a section 2.1.

The materials are very poorly described: the manufacturer and characteristics of hydraulic lime, its setting times and composition are missing. As for the sands, the granulometric curves are not collected, the composition of the sands is not known, their main physical characteristics are not known: water absorption, aggregate density, friability, fines content, particle shape... all these tests are useful, even an XRD of the aggregates is very helpful. Furthermore, the type of water used for mixing must be indicated. 

The test standards are very poorly described, it is advisable to describe the experimental campaign meticulously and even add a photo. EN 12370 refers to salt crystallisation cycles, not freeze-thaw cycles.

The results presented are not discussed and are not compared with other works. 

Line 136 change 3.1.1 by 3.2.1., also, what is the capillary absorption coefficient, it is necessary to indicate this value and only its curve.

Only the mass loss of the samples is presented, no data on flexural strength, hardness and compressive strength of the samples are collected, the loss of strength cannot be evaluated. These are not interesting results for other researchers.

It would be useful to include some microscopy showing how the composition varies with cycling.

In addition, it would also be useful to indicate how the porosity of the samples varies after the tests.

The conclusions are brief and very poor, and should include the many limitations of this research and how the authors intend to remedy them.

The references are not in the journal format.

The translation should be proofread throughout the document

Author Response

The authors have presented a research paper with very little depth in the essays and writing. Unfortunately, I cannot recommend this paper for publication in a scientific journal. I attach some comments justifying the Reject.

Dear Reviewer, thank you kindly for your time and interest in our article, we acknowledge that the first version of the manuscript was quite a draft. We apologize and we completely rewritten most of the sections of the paper, by including your suggestion. The innovation of this paper is both in the application of anti-salt agents in NHL and NHL+ cocciopesto mortars, and the production of dry-mix mortars, instead of using an application of the anti-salt agents by brush over an already hardened mortar, as often found in literature.

[C. Selwitz, E. Doehne, The evaluation of crystallization modifiers for controlling salt damage to limestone, J. Cult. Herit 3 (2002) 205e216. –

Barbara Lubelli, Rob P.J. van Hees, Effectiveness of crystallization inhibitors in preventing salt damage in building materials, Journal of Cultural Heritage, Volume 8, Issue 3, 2007, Pages 223-234, ISSN 1296-2074, https://doi.org/10.1016/j.culher.2007.06.001. -

Ruiz-Agudo, E., Lubelli, B., Sawdy, A. et al. An integrated methodology for salt damage assessment and remediation: the case of San Jerónimo Monastery (Granada, Spain). Environ Earth Sci 63, 1475–1486 (2011). https://doi.org/10.1007/s12665-010-0661-9 and others...]

Line 16 and others, use dots "." as decimal separator.

Dots have been added.

In the abstract it would be convenient to include some significant numerical results.

R: Dear Reviewer, we proceeded to fully rewrite the abstract. Usually, no numeric results are included in the abstract, but we briefly summarize some of the results obtained.

The introduction is very poor, the auotres mention a large number of papers but do not discuss them. It is not a matter of including 56 references in 30 lines, it is a matter of carefully reviewing the literature, highlighting the relevant points of the other papers and highlighting the novelty of the proposed research. This point should be significantly improved.

R: Dear Reviewer, we have improved the argumentation of the introduction, highlighting the most critical aspects on which our research is based on. However, the paper is not a review, so the aim is not to explain each result obtained from others, but to refer to previous work done. We add difference sentences to underline the novelty of the proposed mixtures and the attention also to the fresh state mortar, such as workability and rheology of fresh mortars.

It is recommended that a brief description of the structure and organization of the paper be included after the objective of the paper.

R: done. In section 1. Introduction and in section 2. Materials and Methods.

The materials are very poorly described: the manufacturer and characteristics of hydraulic lime, its setting times and composition are missing. As for the sands, the granulometric curves are not collected, the composition of the sands is not known, their main physical characteristics are not known: water absorption, aggregate density, friability, fines content, particle shape... all these tests are useful, even an XRD of the aggregates is very helpful. Furthermore, the type of water used for mixing must be indicated. 

R: the said corrections have been made, commercial materials used in the experience have been illustrated with specific information and references to technical sheet..

The test standards are very poorly described, it is advisable to describe the experimental campaign meticulously and even add a photo. EN 12370 refers to salt crystallization cycles, not freeze-thaw cycles.

R: we tried to briefly describe the test, but we refer to the norm. Thank you for your warning, we corrected the norm number.

The results presented are not discussed and are not compared with other works. 

R: the results from the tests are presented in detail in section 3.Results and Discussion, further discussion was add and we refer to existing study, however, to best of our knowledge the use of DTMP and Chitosan in NHL mortar has not been reported.

Line 136 change 3.1.1 by 3.2.1., also, what is the capillary absorption coefficient, it is necessary to indicate this value and only its curve.

R: the said corrections have been made.

Only the mass loss of the samples is presented, no data on flexural strength, hardness and compressive strength of the samples are collected, the loss of strength cannot be evaluated. These are not interesting results for other researchers.

R: we add data regarding compressive strength.

It would be useful to include some microscopy showing how the composition varies with cycling. In addition, it would also be useful to indicate how the porosity of the samples varies after the tests.

The microscope results regarding the structures variation were collected, but not significant. The macroscopic observation were more informative. Further pictures have been reported in a supplementary section.

We add the apparent density and total open porosity TOP % values as required on 28days hardened mortars.

The conclusions are brief and very poor, and should include the many limitations of this research and how the authors intend to remedy them.

R: the conclusion section has been rewrited.

The references are not in the journal format.

R: the references have been updated.

Reviewer 2 Report

Abstract is good and structured, explaining the reasons why this research was carried out, but it is necessary to clarify what the aims and benefits of the research are, and it is better if the research method is explained explicitly and clearly.

Introduction, especially in the background section, the author needs to reinforce the arguments and gaps of the research, the formulation of the problem, the urgency of the research, and the aims and benefits of the research.

The research method used is actually good, namely various types of mortar are produced: mortar based on NHL (MS), which is composed by natural hydraulic lime and sand, with a volumetric composition of 1: 2 (lime: sand) and mortar cocciopesto (MSC), where NHL, sand and brick powder are mixed according to the volumetric proportion of 1:1,7:0,3.

The results and discussion are good, where the experimental results show that the physical properties of mortar are affected in different ways by additives, depending on the presence of pozzolanic materials and the additives used. Apart from contributing to meeting the lack of knowledge about the physical properties of traditional NHL-based mortars, the test results turned out to be a good reference for the preparation of new mortars for use in restoration.

The table section needs to be explained properly and sharply so that it is even easier for the reader to understand.

References can be added with updated sources, both books, articles and journals. Thus strengthening the article more academically.

Author Response

AUTHOR 2:

Abstract is good and structured, explaining the reasons why this research was carried out, but it is necessary to clarify what the aims and benefits of the research are, and it is better if the research method is explained explicitly and clearly.

Introduction, especially in the background section, the author needs to reinforce the arguments and gaps of the research, the formulation of the problem, the urgency of the research, and the aims and benefits of the research.

The research method used is actually good, namely various types of mortar are produced: mortar based on NHL (MS), which is composed by natural hydraulic lime and sand, with a volumetric composition of 1: 2 (lime: sand) and mortar cocciopesto (MSC), where NHL, sand and brick powder are mixed according to the volumetric proportion of 1:1,7:0,3.

The results and discussion are good, where the experimental results show that the physical properties of mortar are affected in different ways by additives, depending on the presence of pozzolanic materials and the additives used. Apart from contributing to meeting the lack of knowledge about the physical properties of traditional NHL-based mortars, the test results turned out to be a good reference for the preparation of new mortars for use in restoration.

The table section needs to be explained properly and sharply so that it is even easier for the reader to understand.

References can be added with updated sources, both books, articles and journals. Thus, strengthening the article more academically.

R: Dear Reviewer, we kindly thank you for your kind comments, the corrections about abstract and introductions have been made. We summarized the abstract and add some results. Then, we enlarged the introduction.

We also proceeded to explain better the manuscript tables and checked the references format.

Reviewer 3 Report

This study evaluates the performance of NHL-based mortars for the conservation of historical buildings. The work is in the scope of the journal, however, redaction and structure should be improved as indicated below, especially the methods should be clearer; the author is recommended to identify and practice sophisticated objectives for a journal publication. The author must justify the following points:

Comment 1: The author is using a lot of abbreviations. Hence, it is suggested to include a nomenclature at the beginning of the article.

Comment 2: The abstract should have one sentence per each: context and background, motivation, hypothesis, methods, results, and conclusions.

Comment 3: A deep analysis of recent scientific papers covering only the topic and leading to the submission hypothesis based on the gap analysis of the previously published research is required. Please state the novelty of this paper in answer to the comments: Do we have existing studies on an exactly similar topic? What is the specific contribution relevant to existing similar studies? Please provide some references related to similar existing studies and then the authors could state the contribution clearly in the manuscript.

Comment 4: The proposed approach in section (2) is not outlined with the necessary vigor. The author needs to include sufficient methodological details in the paper and elaborate on the produced results from the proposed methods. Some sections must be added and others need to be relocated and rewritten to make it clearer for the readers. It is not clear to me if this work was built based on a single case study only, or if it was built based on a proposed approach validated within a case study building.

Comment 5: Please state how Figures 1, 2, 3, and 5 were elaborated.

Comment 6: The Results Section should be entitled Results and Discussion. The author needs to understand that this research is not only a case study but is also based on scientific questions. At this level of the study, the author must differentiate between the results of this work based on; the proposed approach; and the applied case study building. This section should be improved by including a clear and concise analysis of all results presented. 

Comment 7: The Conclusion section is missing some necessary details. For example, the author needs to highlight the novelty and the materials and methods used in this work. Then the author should present the results of this work. Eventually, a summary of the limitations of this research as well as a recommendation for future works should be indicated. Please try to avoid using the numerical way in presenting the results.

Comment 8: Overall, the writing style is not up to the mark, the reviewer also is concerned about the contribution of the paper, and findings are inappropriately presented in the abstract. The paper has to go through significant modifications before publication. Besides, do not start with the title and subtitle without a text in between.

Moderate editing of English language required

Author Response

AUTHOR 3:

This study evaluates the performance of NHL-based mortars for the conservation of historical buildings. The work is in the scope of the journal; however, redaction and structure should be improved as indicated below, especially the methods should be clearer; the author is recommended to identify and practice sophisticated objectives for a journal publication. The author must justify the following points:

Comment 1: The author is using a lot of abbreviations. Hence, it is suggested to include a nomenclature at the beginning of the article.

R: Dear Reviewer, the abbreviations before mentioned have been explained in detail in the Materials and Methods section, in which Table.1 describes each formulation in its components and abbreviations.

Comment 2: The abstract should have one sentence per each: context and background, motivation, hypothesis, methods, results, and conclusions.

R: Dear Reviewer, we proceeded to rewrite the abstract.

Comment 3: A deep analysis of recent scientific papers covering only the topic and leading to the submission hypothesis based on the gap analysis of the previously published research is required. Please state the novelty of this paper in answer to the comments: Do we have existing studies on an exactly similar topic? What is the specific contribution relevant to existing similar studies? Please provide some references related to similar existing studies and then the authors could state the contribution clearly in the manuscript.

R: Dear Reviewer, we proceeded to rewrite the introduction section with these useful key-points in mind.

Comment 4: The proposed approach in section (2) is not outlined with the necessary vigor. The author needs to include sufficient methodological details in the paper and elaborate on the produced results from the proposed methods. Some sections must be added and others need to be relocated and rewritten to make it clearer for the readers. It is not clear to me if this work was built based on a single case study only, or if it was built based on a proposed approach validated within a case study building.

R: Dear Reviewer, we proceeded to rewrite the section 2, and provide more useful details about the methodology.

Comment 5: Please state how Figures 1, 2, 3, and 5 were elaborated.

R: Dear Reviewer, the said detail has been added to the manuscript.

Comment 6: The Results Section should be entitled Results and Discussion. The author needs to understand that this research is not only a case study but is also based on scientific questions. At this level of the study, the author must differentiate between the results of this work based on; the proposed approach; and the applied case study building. This section should be improved by including a clear and concise analysis of all results presented. 

R: Dear Reviewer, we proceeded to make the Result and Discussion section more clear and concise, thank you for your precious intervention.

Comment 7: The Conclusion section is missing some necessary details. For example, the author needs to highlight the novelty and the materials and methods used in this work. Then the author should present the results of this work. Eventually, a summary of the limitations of this research as well as a recommendation for future works should be indicated. Please try to avoid using the numerical way in presenting the results.

R: Dear Reviewer, we added the suggested detail in the conclusion.

Comment 8: Overall, the writing style is not up to the mark, the reviewer also is concerned about the contribution of the paper, and findings are inappropriately presented in the abstract. The paper has to go through significant modifications before publication. Besides, do not start with the title and subtitle without a text in between.

R: Dear Reviewer, thank you kindly for your interest in our work, we have rewritten the most critical sections to ensure a better understanding of the paper’s purpose and elaboration of the experimental phase.

Round 2

Reviewer 1 Report

The authors have made multiple corrections to the document. However, the results obtained are still rather poor, and the discussion lacks clarity.

In addition, no microscopic images are included and no chemical characterisation is carried out. 

To some extent, these are the results that have been obtained and cannot be improved. 

The English language should be carefully revised throughout the newly added part.

Author Response

Dear Reviewer, again thank you for your consideration and accurate revisions. This in an explorative work which is analyzing for the first time the possibility of improving the physical resistance to soluble salts and freeze-thaw action of NHL based mortars considered as sustainable materials for the restoration of architectural heritage. The results are pointed voluntarily on the physical and mechanical evaluation, for this is the first step of the research that is able to determine if these formulations are suitable for in situ applications. As a consequence, further research on the chemical effects of addition of DTPMP and Chitosan will be provided. From our revision the discussion seems to be sufficiently clear and exhaustive but if you find them not sufficient or not clear please provide us where they must be improved. About the microscopic images and chemical characterization, as previously said, these weren’t useful in this explorative stage of the research and will be the aim of further papers.

Thank you for your time

Reviewer 3 Report

The author failed to answer several of my previous comments:

I would ask to locate the nomenclature list properly in the manuscript. Table 1 added in the Materials and Methods Section is not the right place to explain the abbreviations.

The author must state how Figures 1, 2, 3, and 5 were elaborated in the proper place in the manuscript. 

The author failed to add the proper discussion of the elaborated results. What is the proposed approach of this work?

The last section should be entitled as Conclusions Section. 

It seem that the Results, Discussions, and Conclusions Sections are confused in this work. 

n/a

Author Response

Dear reviewer, thank you very much for your revisions and suggestions. We are sorry but we believe that the nomenclature table is properly placed as it describes and introduces the different mockups formulations. We moved the information regarding the elaboration of the graphs with Origin 8.5 in the introduction section.

As reported in the introduction the approach of the research is to verify firstly the possibility of enhancing the physical and mechanical characteristics of NHL based mortars, in particular in the presence of freeze thaw events and soluble salts crystallization by the addition of two additives, DTPMP and Chitosan. Since there are few information about the latter and their capacity and the effect on the NHL mortars, our approach is focused on the applicability of these new mixtures and the results showed that these additives slightly enhance the performance of the mortars. At this point we could deepen the study through an evaluation of the additives on the hydration phases and performance through time.

In result and discussion section we reported the results of our tests and commented them, in the conclusion section we provided the most relevant results.

Thank you for your time and consideration.

Round 3

Reviewer 3 Report

The nomenclature is still required to facilitate the understanding issue for the reader. Please note that the nomenclature list must include all the abbreviations used in the research and usually added at the beginning of the manuscript before the Introduction section. 

n/a

Author Response

Dear reviewer, thank you for your suggestions, we added the nomenclature part before the introduction, and checked the language throughout the manuscript.
Future work will be done to investigate the microstructure of the binder by focusing on binder pastes and their maturation with the common methods in use (SEM, XRD, XRF, etc.), the present work is intended as a preliminary investigation on the use of salt-resistance admixtures. Best regards